ecology

endemism, island biogeography, isolation, Last Glacial Maximum, single island endemic, species–area relationship

**Author for correspondence:**
Elisa Barreto
e-mail: elisabpereira@gmail.com

# Area, isolation and climate explain the diversity of mammals on islands worldwide

Elisa Barreto[1,3], Thiago F. Rangel[2], Loïc Pellissier[3,4] and Catherine H. Graham[3]

[1]Programa de pósgraduação em Ecologia e Evolução, and [2]Departamento de Ecologia, Universidade Federal de Goiás, Goiânia, Brazil
[3]Swiss Federal Institute for Forest, Snow and Landscape, Birmensdorf, Switzerland
[4]Landscape Ecology, Department of Environmental Systems Science, Institute of Terrestrial Ecosystems, ETH Zürich, Zürich, Switzerland

EB, 0000-0002-3372-7295; TFR, 0000-0002-2001-7382; LP, 0000-0002-2289-8259;
CHG, 0000-0001-9267-7948

Insular biodiversity is expected to be regulated differently than continental biota, but their determinants remain to be quantified at a global scale. We evaluated the importance of physical, environmental and historical factors on mammal richness and endemism across 5592 islands worldwide. We fitted generalized linear and mixed models to accommodate variation among biogeographic realms and performed analyses separately for bats and non-volants. Richness on islands ranged from one to 234 species, with up to 177 single island endemics. Diversity patterns were most consistently influenced by the islands' physical characteristics. Area positively affected mammal diversity, in particular the number of non-volant endemics. Island isolation, both current and past, was associated with lower richness but greater endemism. Flight capacity modified the relative importance of past versus current isolation, with bats responding more strongly to current and non-volant mammals to past isolation. Biodiversity relationships with environmental factors were idiosyncratic, with a tendency for greater effects sizes with endemism than richness. The historical climatic change was positively associated with endemism. In line with theory, we found that area and isolation were among the strongest drivers of mammalian biodiversity. Our results support the importance of past conditions on current patterns, particularly of non-volant species.

## 1. Introduction

Islands are discrete land areas surrounded by seas. They are distributed all around the world, with broad variation in size, shape, environmental characteristics and degree of isolation [1]. Multiple processes operating over space and time, often influenced by the physical and environmental characteristics of the islands, appear to have resulted in consistent patterns of variation in insular biodiversity [2,3]. However, most work has been conducted on specific islands or archipelagos and there are only a few global evaluations of these emergent patterns (e.g. [4–6]). For mammals, large-scale studies of insular diversity have so far been mostly centred on the island rule (e.g. [7]) and on community structure (e.g. [8,9]), whereas the relationship between biodiversity and characteristics of the island has been limited to certain biogeographic regions and taxonomic groups (e.g. [10–14]). Here we conducted a global study aiming to unveil the generality of the physical and environmental drivers of species richness and endemism of mammals on islands.

The influential model of island biogeography proposed by MacArthur & Wilson [15] posits that island area and isolation influence species richness through the balance between the opposing forces of extinction and immigration.

Lower extinction and higher immigration rates are expected on larger and less isolated islands, as they can support larger population sizes [3,15] and are easier targets for propagules (i.e. target-area effect) that can colonize or repeatedly immigrate to the island (i.e. rescue effect) [16]. Over long time periods, speciation also plays a role in generating insular biodiversity [17]. Speciation rates are greater on larger isolated islands due to greater opportunities for intra-island isolation and the limited gene flow with the mainland or nearby islands [18,19]. Thus, the rate at which new endemic species originate increases with isolation and area because of the greater chance for allopatric speciation in response to low migration and gene flow with the additional chance for intra-island speciation due to area-effects [3,17]. As a result, species richness and endemism should scale positively with area, whereas isolation should have opposing effects, relating positively with endemism but negatively with richness [17].

Islands are not static over time and past geological and climatic conditions left strong imprints on current patterns of biodiversity, especially endemism [20,21]. Physical characteristics of the islands have shifted considerably since the Last Glacial Maximum (LGM), 21 000 years ago. As sea level dropped approximately 122 m, changes in island configuration exposed land bridges that connected the continents to 75% of the world islands larger than 1 km² [1], facilitating biotic and genetic interchange and reducing extinction risk [20,21]. Such increased connectivity during the LGM influenced the mammalian fauna. For instance, in the Japanese archipelago, islands that were connected have greater species numbers and lower endemism than would be expected if they had remained unconnected [11]. Abiotic variability associated with climate change since the LGM is also expected to have influenced current patterns of biodiversity [22]. If insular dynamics are like continental ones, we could expect faster changes in climate over space and time leading to lower endemism because species get extinct or shift its range by tracking climate [22,23].

In addition to the physical characteristics of the islands, insular diversity is also influenced by contemporary environmental conditions. Climate is a major determinant of biodiversity and species number on continental areas and at large spatial scales, with a tendency of increased diversity in warmer, wetter, more productive and climatically more stable environments [24]. Global analysis of island biogeography of plants [5,25], birds [4] and snakes [6] found climatic effects on island biodiversity to be as strong as those reported for continents. However, the comparative role of climate versus physical characteristics of islands on insular diversity patterns is unclear.

Variations in species richness and endemism also arise from the effect of intra-island heterogeneity, in habitat, topography and/or climate, which tends to be greater on larger islands [26,27]. More heterogeneous environments facilitate the coexistence of a wide range of species with different environmental requirements through niche partitioning [28,29]. Additionally, over evolutionary time, greater heterogeneity may increase speciation rates as a result of niche shifts, ecological specialization and increased reproductive isolation [29,30]. Thus, a positive association is expected between topographic and climate heterogeneity and both species richness and endemism.

Drivers of insular biodiversity, especially those associated with intra and inter-island isolation, may depend on species intrinsic characteristics and more specifically on species dispersal ability. For instance, the flight capacity of bats facilitates dispersal over water, essentially creating connections that are not available to other mammals [10]. As a result, biogeographic patterns of bats are substantially different than that of non-volant mammals in continental and insular areas [10,22], where they are the only mammalian group to occupy the largely isolated islands of Hawaii and New Zealand. Hence, the study of insular biota should consider variation in dispersal abilities among clades.

Here we compiled a unique dataset of mammal composition on 5592 islands worldwide to investigate how mammalian richness and endemism relate to island attributes. To obtain insights into evolutionary dynamics [31] we measured insular endemism as the number of single island endemics (SIE) [31,32]. We seek to establish the relative importance of island characteristics as predictors of richness and endemism of bats and non-volant mammals, while accommodating the variation among biogeographic realms due to the deep historical factors [33]. Overall, we expected to find that (i) species richness and endemism are both positively associated with island area, whereas isolation (both, current and past) should be negatively associated with species richness and positively with endemism; (ii) the effects of area and isolation are weaker for bats than for non-volant mammals, and bats have largely overcome limitations imposed by past isolation and, thus, respond more strongly to current isolation; and (iii) the direction of the effect of climate on insular biodiversity is similar to that found on continents, but the effect will be weaker than those of island's physical conditions because area and isolation simultaneously affect the three ultimate causes of biodiversity patterns: speciation, extinction and dispersal.

## 2. Methods

### (a) Mammalian biodiversity data

We used Global Administrative Areas v. 3.6 [34] to subset the spatial polygons of all land masses smaller than Greenland (2 166 000 km²) and that are surrounded by salty water. We overlapped these island polygons with the mammalian range maps from IUCN [35] to derive a global database of mammalian insular biodiversity. We identified a few mismatches in the overlap of island and species polygons that could lead to errors when assigning species presence to islands. These problems were mainly centred in regions with clusters of nearby islands (e.g. Patagonia and Scandinavia) and on islands near the continental shore. We carefully inspected and manually corrected any alignment inconsistencies using QGIS 3.6 [36]. We opted for a highly conservative approach of excluding any island with the slightest doubt about species attribution and ignoring islands where no mammal species occurs according to the IUCN data (i.e. our dataset only includes islands with at least one species).

We removed introduced species from the database by excluding (i) species polygons recorded as introduced by IUCN and (ii) species listed as invasive for each particular island in the Database of Island Invasive Species Eradication [37]. We also removed fully aquatic and marine semi-aquatic species because they are not expected to respond to the characteristics of the islands in the same way as terrestrial species. We ensured that native species that were extinct due to human activity were included in the database by adding occurrence records from [7,38,39].

We used the presence and absence matrix of species per island (doi:10.5061/dryad.hmgqnk9j2) to calculate richness of native species, number of SIE and proportion of SIE per island (figure 1; electronic supplementary material, figures S1–S4 and appendix S2). We contrasted the patterns of mammal biodiversity obtained from IUCN range polygons against published datasets

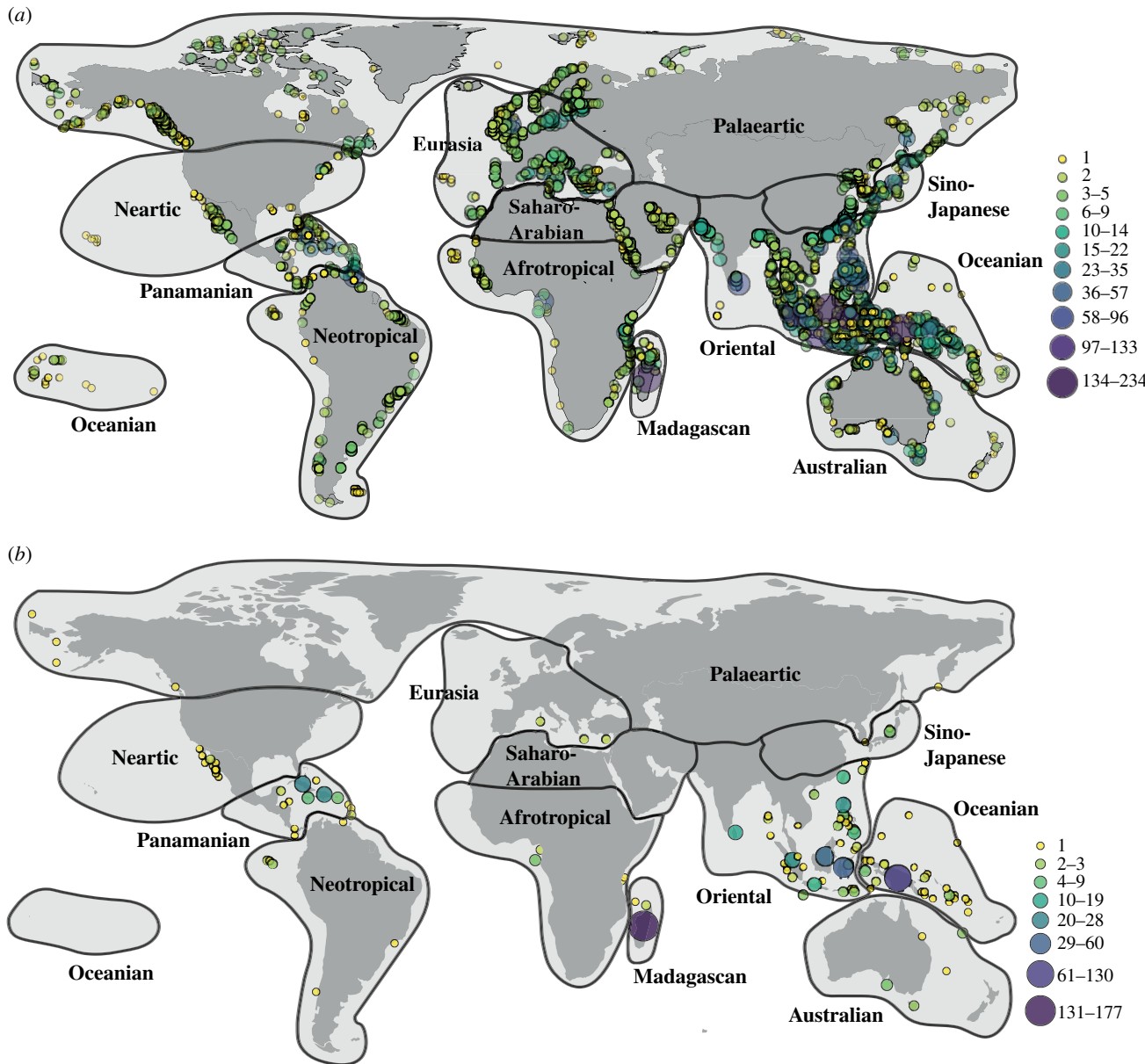

**Figure 1.** (a) Number of mammal species on the 5594 islands with at least one species and (b) number of single island endemics on the 123 islands with at least one endemic. Maps with the diversity of only bats and of only non-volant species are available in electronic supplementary material, figures S1–S4. (Online version in colour.)

that were compiled for specific regions [9,11,13,38,40,41]. The correlation among datasets was high (0.95 ± 0.05), which indicates the reliability of our global dataset. In addition, based on the IUCN range polygons we categorized species as being present only on mainland, on mainland and islands and only on islands. The resulting list was a perfect match to a similar categorization from a mammalian database based on a modification of IUCN range polygons [39], which reinforces the usefulness of our database.

## (b) Physical and environmental characteristics of islands

For each island in our database, we gathered environmental and physical characteristics expected to influence biodiversity: mean annual temperature (in degrees Celsius), annual precipitation (in millimetres) standard deviation of mean annual temperature and precipitation within the island, standard deviation in elevation within the island, area (in km²), surrounding landmass proportion (SLMP), island connectivity to the mainland during the LGM (GMMC) and climate change velocity in temperature since the LGM (CCVT, in metres yr⁻¹) (histograms and the range of values for each predictor are available in electronic

supplementary material, appendix S3 and table S1). We derived temperature and precipitation data from CHELSA using monthly estimates across the years 1979–2013 [42] and elevation from the Global Digital Elevation Model GTOPO30 [43] and calculated mean and standard deviation per island using QGIS 3.6 [36].

We obtained island area, SLMP, GMMC and CCVT from a public island characterization database [1] by matching the centroid coordinates to the island polygons. SLMP is a proxy of island isolation with great predictive power and was calculated as the $\log_{10}$-transformed sum of the proportion of surrounding landmass within buffer distances of 100, 1000 and 10 000 km around each island perimeter [44]. GMMC is a binary descriptor of historical isolation that uses past and present global bathymetry data to infer if islands were connected to the continent during the LGM by assuming the estimated sea level decrease of 122 m at 18 000 years ago (more details in [1]). We multiplied SLMP by −1 and coded GMMC as 0 being connected and 1 being disconnected to the mainland during the LGM, so both metrics represent isolation (i.e. higher SLMP and GMMC represent greater isolation). Hereafter we will refer to those variables as 'current isolation' and 'past isolation'. CCVT over the past

21 000 years was calculated by dividing the difference in mean annual temperature between past and present by the spatial change in present mean temperature [1,45]. CCVT is interpreted as the speed at which the organism would have to move to keep pace with historical temperature change, assuming no change in topography [1,45].

Islands were classified into the 12 global mammalian zoogeographical regions [33] (figure 1), hereafter 'realm'. We removed 505 islands from the dataset because it was not possible to derive all environmental variables or to assign a realm with confidence, usually because they were small (less than 1 km$^2$) or located on a biogeographic boundary. The final dataset comprised 5592 islands (out of the approx. 17 000 islands larger than 1 km$^2$ worldwide [1]) of which 123 contained SIE (figure 1; electronic supplementary material, appendix S2).

## (c) Data analysis

We standardized all quantitative predictors (mean = 0 and standard deviation = 1) to enable comparison among regression coefficients and we transformed those that were non-normally distributed by using natural logarithm base 10 or square root to reduce asymmetry. Multicollinearity among predictors was calculated with the variance inflation factor (VIF) and was generally low (mean 1.86 ± 1.06 s.d.; electronic supplementary material, table S2). Standard deviation in precipitation had high VIF values (approx. 7) but was maintained in the final models because its exclusion did not change the results. To explore broad biodiversity patterns, we tested if islands that harbour only bats (n = 1831), only non-volant species (n = 2094) or representatives from both groups (n = 1667), tend to differ in latitude, physical and environmental characteristics using ANOVA tests (electronic supplementary material, figures S5 and S6).

We related the biodiversity of the islands to their physical and environmental characteristics by fitting generalized linear models (GLMs) to the number of SIE and generalized linear mixed-effect models (GLMMs) to species richness and proportion of SIE (pSIE) using the packages glmmTMB [46], lme4 [47] and MASS [48] in R. SIE was modelled using GLM because most islands contain only one or two endemics, and therefore there was not enough variation in this variable to fit a GLMM. Species richness and SIE were modelled using a negative binomial error distribution and pSIE was modelled using a binomial error distribution with species richness used as prior weights. Only islands with at least one single island endemic were included in the models fitted for endemism (n = 123). Analyses were conducted for all mammals and separately for bats and non-volant species. Results for pSIE did not result in additional insight than the models for species richness and SIE and are thus only presented in electronic supplementary material, figure S7. As our dataset contains no island with zero species, we subtracted one (1) from the species richness and from the number of endemics when modelling the biodiversity of all mammal species to improve the fit of negative binomial models because these models are designed to predict zeros. This approach yielded similar results as that of a zero-truncated model, which is only available for GLMMs.

We fitted GLMMs for species richness using realm as random effect to enable the estimation of different intercept and slopes for each realm across all the predictors, as this reduces type I error when compared to models with only random intercept [49]. Inclusion of realms as random effect accommodated the regional differences that are expected due to historical factors [33,50,51] and that might cause spatial autocorrelation at regional scales [52].

None of the models had zero inflation or overdispersion, which was tested by simulating standardized residuals from the fitted models in DHARMa R package [53]. We estimated spatial autocorrelation in the residuals using Moran's I correlogram based on the geodesic distances among islands in the

software SAM [54] and it was only detected at spatial scales so small it is unlikely to bias p-values (electronic supplementary material, tables S3–S11 and figure S8). Conditional and marginal pseudo coefficients of determination ($R^2$) were calculated following Nakagawa et al. [55] using MuMIn [56] R package. In the main text, we present the more conservative pseudo-$R^2$ estimates (i.e. trigamma for GLMM's and delta for GLM's). All other $R^2$ estimates can be found in electronic supplementary material, tables S12–S14.

# 3. Results

## (a) Diversity patterns

Mammalian richness on islands ranged from one to 234 species (islands with no species were excluded from the dataset), with most islands being home to only one (42.9%) or two (20.8%) species (figure 1a). New Guinea, Borneo, Madagascar, Sumatra, Sulawesi and Java had the richest mammalian faunas, each containing more than 100 species (figure 1a). The richest fauna of bats was found in Borneo (92 species; electronic supplementary material, figure S1), whereas Madagascar and New Guinea hosted the richest faunas of non-volant mammals (169 and 167 species, respectively; electronic supplementary material, figure S2). We identified 782 species that are endemics to a single island, most of which are non-volant species (86.1%). Madagascar and New Guinea had the largest number of endemics, which respectively accounted for 87.6% and 55.5% of their mammalian fauna (figure 1b; electronic supplementary material, figures S3 and S4). As found for species richness, most islands with endemics hosted only one (63.6%) or two (14.4%) SIE (figure 1b).

We tested whether islands that harbour only bats, only non-volant mammals or both, differ in their physical and environmental characteristics (figure 2; electronic supplementary material, figures S5 and S6). Compared to islands where either bats or non-volant mammals occur, islands where both groups co-occur were larger, less isolated and had greater spatial variation in environmental conditions (figure 2; electronic supplementary material, figure S5). Islands occupied only by bats had the opposite characteristics; they were considerably more isolated and were also warmer and wetter (figure 2; electronic supplementary material, figure S5). Islands where only non-volant mammals occur have experienced significantly greater climate change velocity since the LGM, were colder and located at higher latitudes than those islands that supports only bats or both (figure 2; electronic supplementary material, figure S5). Differences among islands regarding the occurrence of SIE were largely similar to that of species richness (electronic supplementary material, figure S6).

## (b) Drivers of island diversity

Our models explained between 27% and 94% of the variation in island biodiversity, with species richness being better explained (mean $R^2 = 0.89 \pm 0.06$ s.d.; table 1) than endemism (mean $R^2 = 0.55 \pm 0.22$ s.d.; table 1). The mixed model fitted for richness revealed that a large proportion of the explained variation is attributed to the variance among biogeographic realms (comparison of marginal versus conditional $R^2$ in table 1). Overall, we found that species richness was associated more strongly to area (+), current and past isolation (−) and mean temperature (+ for bats), whereas the other predictors

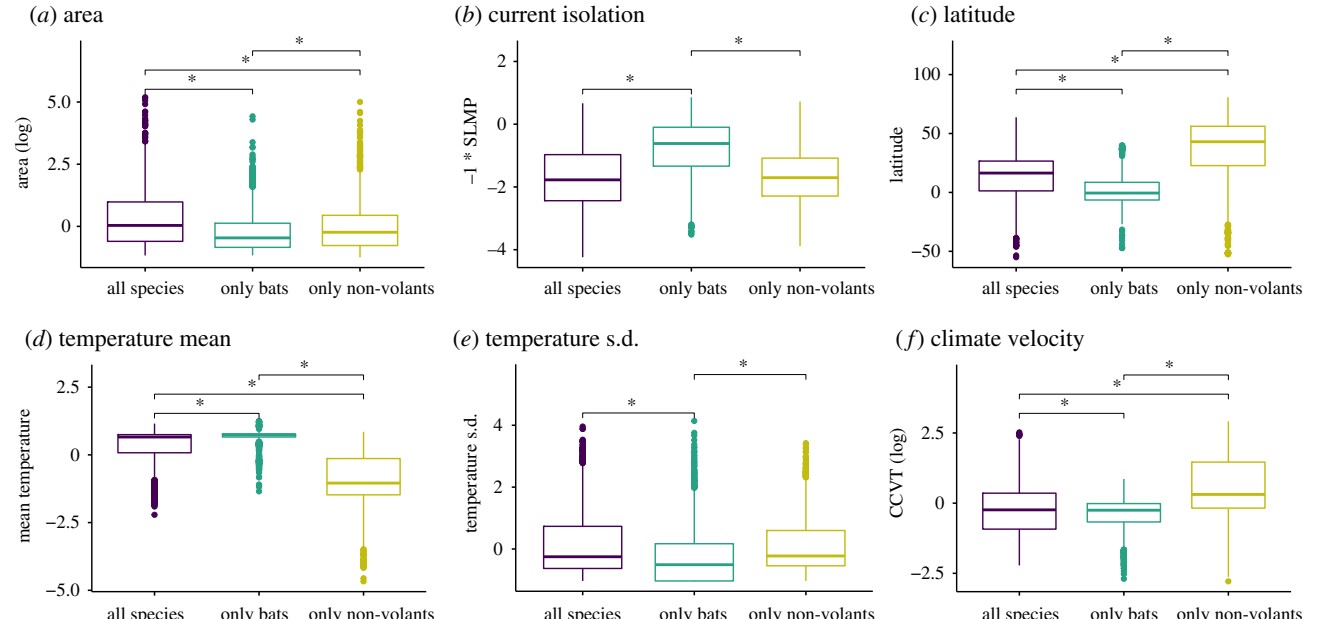

**Figure 2.** Comparison among islands that harbour only bats ($n = 1831$), only non-volant mammals ($n = 2094$) or both ($n = 1667$), regarding their (*a,b*) physical, (*c*) latitude and (*d*–*f*) environmental characteristics using ANOVA tests. Asterisks represent $p \leq 0.05$ between pairwise comparisons. All variables except latitude are standardized to mean = 0 and s.d. = 1. SLMP, surrounding land mass proportion. CCVT, climate change velocity. ANOVA results for all predictors available in electronic supplementary material, figures S5 and S6. (Online version in colour.)

**Table 1.** Pseudo coefficients of determination (pseudo-$R^2$) of the mixed model used to model species richness and the generalized linear models used to model the number of single island endemics of all mammals and of bats and non-volants separately. In mixed models, the marginal $R^2$ describes the proportion of the variance in biodiversity that can be explained only by the physical and environmental characteristics of the islands (i.e. fixed factors), and the conditional $R^2$ describes the proportion explained by the entire model, including fixed factors and the realm (i.e. random factor).

| | species richness | | |
| --- | --- | --- | --- |
| | conditional | marginal | endemism |
| all mammals | 0.83 | 0.19 | 0.68 |
| bats | 0.94 | 0.21 | 0.30 |
| non-volant | 0.90 | 0.19 | 0.68 |
| mean | 0.89 | 0.20 | 0.55 |

had comparatively weak or no relationship with richness (figure 3*a*). By contrast, we found that endemism was strongly related to a more varied set of predictors, including climate velocity (+) and mean precipitation (−), for example (figure 3*b*).

### (i) Area and isolation

Area and isolation had the strongest (figure 3) and most consistent relationships with species richness across biogeographic realms (figure 4). As expected, island area was associated with increases in species richness and endemism (figure 3). Also, islands that are currently isolated, or have been isolated in the past, had a negative relationship with richness and positive relationship with endemism (figure 3).

Contrary to our expectations, the effects of area and isolation on bats were not always weaker than for non-volants, only for the endemism–area relationship. Area had a similar effect on endemism ($\beta = 0.51 \pm 0.2$ s.e.) and on richness of bats

($\beta = 0.54 \pm 0.13$ s.e.), whereas among non-volants, area had a twice as strong effect size in the relationship with endemism ($\beta = 1.23 \pm 0.18$ s.e.) than with richness ($\beta = 0.69 \pm 0.12$ s.e. for all mammals and $\beta = 0.63 \pm 0.13$ s.e. for non-volants). The relationship between area and richness of all mammals and of bats was significantly greater the more isolated the island (figure 3*a*, interaction term). As predicted, flight capacity influenced the relative importance of past versus current isolation, as the diversity of bats tended to be more strongly related to current isolation and past isolation had no significant effect (figure 3). Conversely, richness of non-volants was more strongly related to past than current isolation (figure 3), as expected.

### (ii) Environmental conditions

Environmental factors emerged as stronger predictors of endemism than of richness (figure 3). Smaller environmental–richness slopes resulted partially from the variation in strength and in direction of these relationships among biogeographic realms (figure 4). For example, richness of all mammals had a strong negative relationship with temperature in the Afrotropics, but a positive one in Oceania (figure 4*a*). Variation in temperature within the island was the environmental factor with the most consistent direction of effect across realms, being positively associated with richness, but with a small effect size (figure 4). We found mean precipitation to relate negatively with endemism, whereas variation in precipitation within the island had a positive relationship with endemism of bats (figure 3). Endemism had a positive association with climate change velocity and a negative association with topographic variation (figure 3*b*).

## 4. Discussion

Global patterns of biodiversity of insular mammals are well explained by the physical, biogeographic and environmental

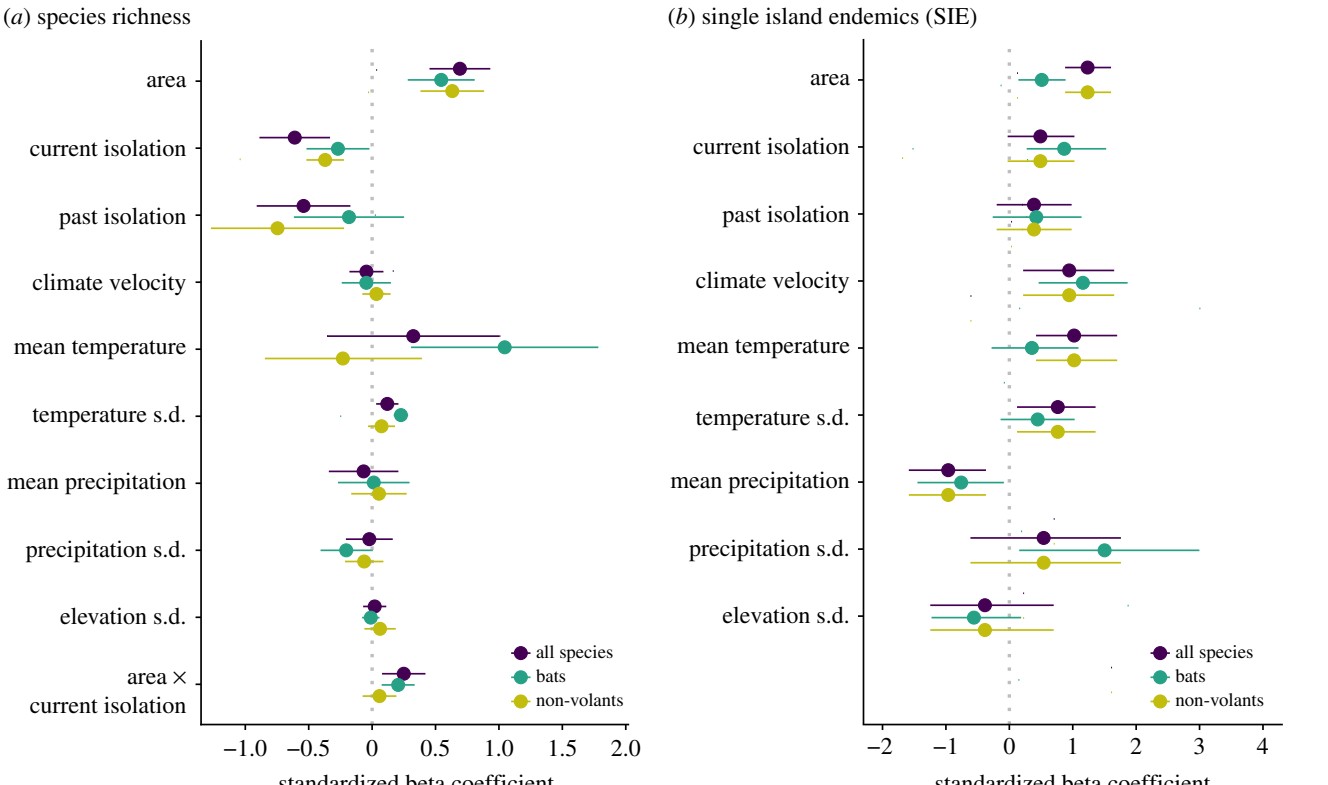

**Figure 3.** Standardized beta coefficients of (a) the mixed models used to model species richness and (b) the generalized linear models used to model endemism (number of single island endemics) of all mammals and of bats and non-volants separately. Dots indicate estimated standardized beta coefficients and error bars represent 95% confidence intervals. Beta coefficients, standard errors and z-values are available in electronic supplementary material, tables S15 and S16. (Online version in colour.)

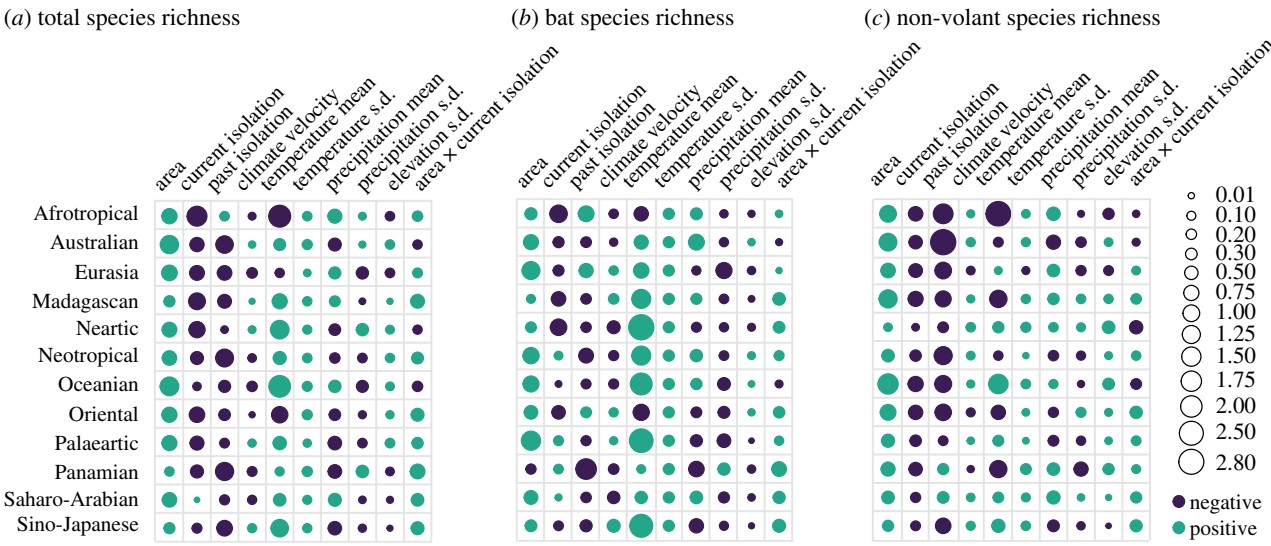

**Figure 4.** Standardized beta coefficients per biogeographic realm for species richness of (a) all mammals, (b) bats and (c) non-volant species estimated from the generalized linear mixed models with biogeographic realm as a random factor. Dot size represents effect size and colour represents the direction of effect. Coefficients are available in electronic supplementary material, tables S18–S20. (Online version in colour.)

characteristics of the islands. In line with the theory of island biogeography and its further developments [15,17,31], both species richness and endemism increased with area, whereas endemism increased but richness decreased with isolation. Environmental conditions also played a role in shaping insular biodiversity, with varying effects across realms and stronger effects on endemism. Overall, species richness was better explained than endemism (mean $R^2$ of 89% versus 55%, respectively; table 1) because we were able to

statistically accommodate biogeographic realms as a random effect. This result suggests a strong role of historical contingency as a source of variation in insular biodiversity. Also, endemism is likely to be strongly influenced by island age [31] and intra-archipelagic dynamics [57], information that is still unknown for most islands. The relative importance of physical characteristics differed between bats and non-volants given their difference in dispersal ability. Bats have largely overcome the effects of past island isolation

and are also less affected by island area, at least for endemism. By taking a global view of the drivers of insular biodiversity, we show the universality of the effects of area and isolation, whereas environmental conditions, which are ubiquitous drivers of richness on continents [24], have variable influence on island biodiversity.

We acknowledge that the study of current insular biodiversity patterns could potentially be impacted by the recent extinctions and range shifts due to human activities (e.g. [38,58]) and to the Linnean and Wallacean shortfalls [59]. For instance, a recent study shows that half of the mammal species on the well-known island of Luzon were unknown only a few years ago [60]. Despite these limitations, our findings are in line with the various expectations from island biogeography theory. That suggests that IUCN-derived diversity measures are robust to explore patterns of insular biodiversity [8], especially because of the large spatial scales and great number of sampling unities that can be derived from such dataset.

## (a) Effects of area and isolation

Mammals proved to be a textbook example of how island area is positively associated with species richness and endemism, whereas isolation is negatively related to richness, but positively with endemism at a global scale [2,15]. The positive species–area and endemism–area relationships comes close to being a universal law of ecology [61,62] and is well documented for mammals at smaller scales, on both true islands (e.g. [12,56]) and mountain tops (e.g. [57,58]). The strong effect we found for area is unlikely to arise from a confounding effect of habitat heterogeneity [26], given that we recovered negative or weak relationships between within-island elevation and diversity at the global scale. The positive species–area association holds across realms (except for the richness of bats in the Panamanian), unlike relationships observed for other predictors, and thus strongly support the generality of the positive species–area relationship, probably due to increases in speciation and immigration rates and decreases in extinction rates [3].

The contrasting relationships of island isolation with richness (negative) and endemism (positive) are consistent with the expected reduction in immigration and gene flow in more isolated islands [3,16,19,32] so that increases in diversity are mostly due to endemic species [17,32]. Measures of island isolation at global scale should not focus solely on the distance to the mainland, but ideally, incorporate elements of the landscape, such as stepping stone distances or information on the surrounding landmasses [44]. This is especially the case for mammals, whose neighbouring islands are known to be a more important source of colonizers than the closest mainland [11]. We were able to capture the strong effects of isolation by using a measure based on the proportion of surrounding landmass [44], giving support to the growing body of evidence of the importance of archipelago configuration and spatial structure of islands as drivers of biodiversity [57,63].

## (b) Flight capacity

The strength of the relationship between area and isolation with biodiversity varied with the group's vagility and the type of diversity measured and is likely to reflect the relative importance of different processes, namely dispersal,

extinction and speciation [62]. The flight capacity of bats enables them to move more easily between and within islands and mainland [64], thus maintaining effective population rescue that decreases rates of extinction, but hinders speciation events [17,19]. Accordingly, we found that bats had weaker endemism–area association than non-volants, suggesting that intra-island diversification is weaker on organisms with greater vagility. Moreover, island area had a stronger relationship with the number of SIE than with the richness of non-volants, as expected from increased *in situ* speciation resulting from a weaker rescue effect.

The relative importance of past versus current island isolation among bats and non-volants also points to differences in vagility driving global spatial patterns of insular biodiversity. Non-volant mammals still carry imprints of past isolation, whereas bats have largely overcome such historical effects and are more strongly associated with the current than past isolation. This finding contrasts with that of phylogenetic endemism of mammals across the mainland and large islands, for which past isolation was a stronger predictor even among bats [22]. Such contrasting result could be that phylogenetic patterns are better at capturing the long-lasting effects of historical events than are patterns based on taxonomic diversity.

The greater dispersal capacity of bats over water probably resulted in their sole occupancy of smaller, more isolated islands that have narrow spatial variation in environmental conditions. Bats are more common on warmer and wetter islands, as expected given their tropical origin and strong niche conservatism [65]. Few Chiropteran lineages have been able to overcome the energy constraints imposed by low temperatures limiting their ability to colonize temperate regions and high elevations [66,67], explaining why we found temperature effects on bat richness to be greatest in colder regions, such as the islands in the Nearctic and Palaeartic realms.

## (c) Effects of the environment

Environmental factors, such as climate, are often good predictors of global insular biodiversity across different groups (e.g. birds [4], plants [5,57], snakes [6] and even human languages [68]). For mammals, we found that overall, physical conditions of the island (i.e. area and isolation) are stronger predictors of richness than environmental conditions, probably because they influence all three processes: speciation, extinction and immigration. Environmental factors are more strongly associated with endemism than with richness and the intensity and direction of the effect varied considerably across realms suggesting that environment–diversity relationships on islands are context-dependent and contingent to regional/archipelagic historical effects [57]. Specifically, variation in environmental–diversity relationships could be the result of different mechanisms prevailing in each realm or of the specific adaptations that each species pool acquired by evolving under different environmental gradients.

Increased heterogeneity in precipitation and temperature within islands tended to relate to greater endemism, potentially because greater variation in climate might facilitate niche partitioning and reproductive isolation via ecological specialization [29,30]. However, the same did not hold for elevation heterogeneity. We argue that at such large spatial scales, the effect of elevation that could be detected in insular biodiversity mostly results from the environmental conditions

that covary with it. In addition, elevation measures at finer resolutions are likely to be needed to detect the strong effects of elevation that are often found on islands (e.g. [26,63]).

The positive association between endemism and climate velocity since the LGM contrasts with findings for endemism on the mainland [23] and challenges the expectation that climate instability would favour the occurrence of generalist broad-ranged species [69]. On the continent, changes in temperature since the LGM influenced mammalian bio-diversity through range shifts and increased the probability of extinction [69]. On islands, where boundaries are hard, species may be less likely to shift their ranges to track new climates and may be under strong pressure to adapt to novel conditions. Adaptation of island organisms can quickly lead to divergence from populations of other islands and mainland [70] and to speciation events that culminate in an increase in SIE. Also, endemism might increase with climatic instability if environmental changes were com-paratively milder on the focal island than on the mainland, which could be the case given that the ocean mass buffers climatic change on islands [71]. In this case, species that went extinct on the continent might have persisted on the island.

## 5. Conclusion

Islands provide a unique opportunity to parse out the impor-tance of different mechanisms that generate and maintain diversity. On one hand, we find that area and isolation, both past and present, strongly and consistently relate to mammalian richness and endemism globally. The relative importance of these drivers was consistently associated with species vagility, a result that can prove useful in other

contexts such as in restoration systems and conservation efforts. On the other hand, climatic conditions have more idiosyncratic relationships with island diversity globally that suggests a variation in the main processes taking place at smaller spatial and taxonomic scales that would benefit from a more in-depth regional or clade-specific focus.

Data accessibility. The full dataset used in this publication is available in the electronic supplementary material [72] and from the Dryad Digital Repository: https://doi.org/10.5061/dryad.hmgqnk9j2 [73] (the presence and absence matrix is under a 2-year embargo).

Authors' contributions. E.B.: conceptualization, data curation, formal analysis, investigation, methodology, visualization, writing—original draft, writing—review and editing; T.F.R.: conceptualization, funding acquisition, methodology, supervision, writing—review and editing; L.P.: conceptualization, methodology, writing—review and editing; C.H.G.: conceptualization, funding acquisition, methodology, supervision, writing—review and editing. All authors gave final approval for publication and agreed to be held accountable for the work performed therein.

Competing interests. We declare we have no competing interests.

Funding. This study was financed in part by the Coordenação de Aper-feiçoamento de Pessoal de Nível Superior—Brasil (CAPES)—Finance Code 001; the Swiss Federal Institute for Forest, Snow and Landscape (WSL); and the MCTIC/CNPq (grant no. 465610/2014-5) and FAPEG (grant no. 201810267000023) in the context of the National Institute of Science and Technology (INCT) in Ecology, Evolution and Biodiversity Conservation. E.B. was supported by a doctorate and a 'sandwich' fellowship from CAPES and a CNPq/DTI-A Fel-lowship from INCT. E.B. and C.H.G. acknowledge funding support from the European Research Council (ERC) under the European Union's Horizon 2020 research and innovation programme (grant agreement no. 787638) granted to C.H.G.

Acknowledgements. We are thankful to Ana Santos, Matthew Helmus, Katherine Hébert and Jean-Philippe Lessard for sharing data to be compared against ours.

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
