## [Peer Review File · Proceedings of the Royal Society B: Biological Sciences]

Review History

RSPB-2021-1879.R0 (Original submission)

Review form: Reviewer 1

Recommendation

Major revision is needed (please make suggestions in comments)

Scientific importance: Is the manuscript an original and important contribution to its field?

Good

General interest: Is the paper of sufficient general interest?

Excellent

Quality of the paper: Is the overall quality of the paper suitable?

Good

Is the length of the paper justified?

Yes

Should the paper be seen by a specialist statistical reviewer?

No

Do you have any concerns about statistical analyses in this paper? If so, please specify them explicitly in your report.

Yes

It is a condition of publication that authors make their supporting data, code and materials available - either as supplementary material or hosted in an external repository. Please rate, if applicable, the supporting data on the following criteria.

Is it accessible?

Yes

Is it clear?

Yes

Is it adequate?

Yes

Do you have any ethical concerns with this paper?

Yes

Comments to the Author

General comments:

I have read the manuscript titled 'Area, isolation, and climate explain the diversity of mammals on island worldwide'. In this paper, the authors tested several key predictions of the island biogeography theory using mammal data worldwide. They found contrasting patterns between the insular diversity of bats and non-volant mammals and confirmed the leading role of island area and isolation in driving island biodiversity.

The paper was written in a succinct and logical manner. The results and the arguments are easy to follow. In general, I think the paper makes a nice contribution to the knowledge of island biodiversity worldwide. I have three general comments to further improve the paper:

1. I think the theoretical background of the study should be further developed in the introduction, especially regarding how pSIE is predicted by extinction and colonization rates. There is a nice theoretical paper on this subject, which I think the manuscript could benefit a great deal from:

Chen, X.-Y., & He, F. (2009). Speciation and endemism under the model of island biogeography. *Ecology*, 90(1), 39–45. doi:10.1890/08-1520.1

It might provide additional insights about why island area does not explain the pSIE (Fig. 3). This deviation from your prediction should also be given more attention in the discussion section.

2. I wonder if the current title is a little bit misleading and masks some important messages unique to the paper. For example, judging from table 2, biological realm explains much more variation of insular biodiversity than any other variables. This seems to suggest the dominant role of historical contingency/evolutionary history on island biodiversity. The current method treats the biological realm as random effects. However, I am curious to know if each biological realm is fitted by a different model with the same set of predictors, will the R² be different among biological realms? In other words, will climatic and insular variables have different explanatory powers in different regions?

3. More introductions should be provided addressing the prior expectations about the relative importance of climatic variables and island variables. In addition, the effects of climate

change velocity and elevation heterogeneity should also be properly introduced before the method section.

Minor comments:

1. L61, lower extinction rate on larger island could also contribute to higher pSIE (Equation 5; Chen and He, 2009). But if the island species richness is fixed, lower extinction rate (larger islands) can actually lead to lower pSIE (Fig. 3; Chen and He, 2009)

Chen, X.-Y., & He, F. (2009). Speciation and endemism under the model of island biogeography. *Ecology*, 90(1), 39–45. doi:10.1890/08-1520.1

2. L82, 'and species numbers' is redundant.

3. L103, is there a reason that climatic effect is expected to be weaker than island area and isolation?

4. L186, not enough variation in what variable? What would be the random effect if a glmm is used?

5. L186-188, since pSIE is a proportion data. I think beta regression might be more suitable for this. The authors might consider doing a beta regression and put it in the supplementary (but this is totally up to the authors).

6. L191-193, I am not familiar with this approach. Could you give more explanations about why subtracting 1 from species richness improves the fitting of a negative binomial model? Also please add citations if this has been implemented in other studies.

7. L196-200, could you be more specific about which predictors have biological realm as a random effect for the slope (all the predictors or just some of them?).

8. L214, how many islands were excluded from the analysis? It should be mentioned in the method section.

9. Figure 1, 'ate least' should be 'at least'.

10. Figure 2, all species include bats and non-volant species. I suggest removing the 'all species' group as it is not independent of the other groups, a pairwise comparison in this case is therefore not appropriate.

11. L263, the tables start with table 2. There is no table 1 in the paper.

12. L263, table 2. Consider adding the results of the partitioning of variance of climatic variables and island variables.

13. L341, it is worthwhile to explain why area does not influence pSIE, which is contradictory to the prediction of Chen and He 2009.

14. Consider removing some tables in S2-S9. It seems unnecessary to include all the spatial-autocorrelation analysis.

Review form: Reviewer 2

Recommendation

Accept with minor revision (please list in comments)

Scientific importance: Is the manuscript an original and important contribution to its field?

Good

General interest: Is the paper of sufficient general interest?

Excellent

Quality of the paper: Is the overall quality of the paper suitable?

Excellent

Is the length of the paper justified?

Yes

Should the paper be seen by a specialist statistical reviewer?

No

Do you have any concerns about statistical analyses in this paper? If so, please specify them explicitly in your report.

No

It is a condition of publication that authors make their supporting data, code and materials available - either as supplementary material or hosted in an external repository. Please rate, if applicable, the supporting data on the following criteria.

Is it accessible?

Yes

Is it clear?

Yes

Is it adequate?

N/A

Do you have any ethical concerns with this paper?

No

Comments to the Author

In this manuscript, the authors examine the drivers of native mammal richness and endemism in islands globally. They perform analyses for all mammals, for bats only and for non-volant mammals. Geographical features (area and isolation) explain most of richness and endemism, whereas environmental drivers appear to be more idiosyncratic and their effects vary across biogeographic realms.

GENERAL COMMENTS

This is a nice, pretty straightforward paper that present convincing results. The manuscript covers all major pitfalls I could think of, in terms of data compilation and analyses. I only have some minor comments described below to clarify a few points.

The data available in Supplementary Information only contains the summary data per island (richness, SIE and pSIE), rather than the full list of species. It would be extremely useful to the scientific community to make the whole data available in a public repository, with a DOI so that it is citable and credit can be given to the authors of course.

SPECIFIC COMMENTS

l.31: Why is this result unexpected? Actually, important changes in climatic conditions might lead to speciation through adaptation, so it does not seem counterintuitive.

l.34: evidence on → evidence of

l.47: the full stop should be a comma, or the new sentence should be rephrased to be grammatically correct.

l.61: the relationship between area and endemism should be unpacked a bit. If I get the reasoning correctly, if a population gets isolated from the continent, it can evolve in a new species due to

allopatric speciation, regardless of area, right? But the larger the island, the more opportunity for additional speciation events to occur, leading to more than potentially one new species?

l.149-151: which time period was used to compute the temperature and precipitation means and sd?

l.162: I do not get the meaning of the sentence “Due to the lower number of sample units per realm on the analyzes of pSIE”. By “sample units”, do you mean islands? Why are there less sample units for pSIE than for richness and SIE?

l.171: would it be possible to get the commented R script used for the analyses as supplementary information?

l.238: according to figure 2e, temperature variability was only significant when comparing all species and bats, not all species and non-volant species.

l.234-244: this section only refers to significance, but some differences are larger than others, and it would be good to mention effect size. For example, distributions were significantly different for area and temperature sd, but the differences were quite small compared to current isolation, for example.

Table 2 should be Table 1

l.251: According to Table 2, the range of conditional R2 is between 27% and 94%, not 27% and 90%.

l.259: Figs. 2a → Fig. 3a

l.291-294: I do not think you can directly compare absolute values of coefficients between models, since the response variables differ.

l.296: why was the interaction between area and isolation only used for richness, and not the other 2 models?

Figure 4: It would be useful to distinguish between significant and non-significant coefficients. Significant coefficients could be represented by filled disks, and non-significant coefficients by hollow rings, using the same colours for positive and negative values.

l.306: the colours in Figure 4 are not blue and red.

l.309-321: It could be useful to show the range of values for environmental variables in the different realms, and to discuss if similar trends in coefficient values can be associated with similar range of values (e.g. possibly positive and negative relationships for realms with lower and higher temperatures, respectively). Also, there could be interactions between environmental variables. For example, the relationship between richness and temperature may differ in dry and wet environment?

l.330: then → than

l.330: where do these values come from? There do not appear in Table 2.

l.340: The effects of area and isolation → Effects of area and isolation

l.358: “elements of the landscape” is unclear. Please be more specific, and give a couple of examples.

l.366: The flight capacity → Flight capacity

l.369: Name these different processes. I guess dispersal, environmental and speciation processes?

l.395: The effects of the environment → Effects of the environment

l.396 onwards: This section lacks some discussion on why the effects of climate vary so much with the realm.

l.410: is mostly from → mostly results from

l.411: And that → In addition,

l.412: it is needed elevation measures at finer resolutions → elevation measures at finer resolutions are needed.

l.414-422: I am not sure if the last explanation really makes sense here, as according to it, to be endemic in an island with high VCC, this one island must still have a lower VCC than multiple surrounding ones in which the same species would go locally extinct. It does not seem that this could generate a positive relationship. This explanation could also be tested by using the difference of VCC in an island with the VCC in surrounding islands. Rather, my guess is that in the mainland, vagile mammal species can move to follow climate change, whereas they cannot on islands. To survive, they would need to adapt, which can lead to speciation.

l.429: associated to species attributes (i.e. flight capacity) → associated with species vagility/dispersal ability/dispersal mode. "Attributes" is too general, whereas is specifically refers to a specific attribute.

l.434: "For instance" does not really work here. It suggests an example of how using "more in-depth regional or clade-specific focus" would enable us to better understand the relationship between endemism and environmental drivers will follow, but instead moves on to generalize the insights from this study to other systems.

l.436: The link between this study and deforestation, which would focus on habitat loss and exclude speciation, is not clear and seems a bit far-fetched.

Review form: Reviewer 3

Recommendation

Accept with minor revision (please list in comments)

Scientific importance: Is the manuscript an original and important contribution to its field?

Excellent

General interest: Is the paper of sufficient general interest?

Excellent

Quality of the paper: Is the overall quality of the paper suitable?

Excellent

Is the length of the paper justified?

Yes

Should the paper be seen by a specialist statistical reviewer?

No

Do you have any concerns about statistical analyses in this paper? If so, please specify them explicitly in your report.

No

It is a condition of publication that authors make their supporting data, code and materials available - either as supplementary material or hosted in an external repository. Please rate, if applicable, the supporting data on the following criteria.

Is it accessible?

Yes

Is it clear?

Yes

Is it adequate?

Yes

Do you have any ethical concerns with this paper?

No

Comments to the Author

This is an ambitious attempt to uncover generalities in the drivers of mammal diversity on islands at the global scale. The authors built an extensive global dataset of mammal diversity on islands, and compiled an accompanying dataset of physical and climatic characteristics of these islands to test hypotheses grounded in island biogeography theory. They succeed in uncovering generalities in the drivers of diversity that have not yet been identified for mammals on islands at the global scale. Overall, the study presents these contributions in a way to interest a broad readership as well as readers with a specific interest in island biogeography and mammals.

This is an important contribution to the body of evidence supporting the importance of island area and isolation in shaping diversity. The dataset which was compiled for this study is also a valuable contribution to macro-ecological research, due to its scale, the inclusion of bats and endemic species, and the rigour with which it was built. The methodology for data collection, validation, and analysis is justified both ecologically and statistically throughout the study. The manuscript was well-written and had a good logical flow throughout the text, and was overall an interesting read. The introduction sets up the study on a solid foundation of theoretical concepts that are grounded in examples and knowledge of how mammals live on and disperse between islands. I have just a few questions and suggestions about points that deserve more clarification or elaboration to drive the study's message home.

My main comment is about endemism, which is measured both as the number of single island endemics (SIE) and as the proportion of single island endemics (pSIE). I found myself wondering why both of these measures were investigated rather than just pSIE or vice-versa. What was the reasoning for including both? It could be helpful to explain this decision early in the text, especially if it is because they give different insights. The two metrics were also related quite differently to isolation in the results, but the reason for this should be discussed somewhere to guide the interpretation of this result. If there is not a clear justification for keeping both, I would consider focusing on one of the two (maybe pSIE) in the main text to have a single metric for richness and a single metric for endemism, which would simplify the manuscript.

The realm labels in Figure 1b) could be larger. They are difficult to read, especially on a smaller screen.

Line 181: I believe this reference to Figs S1 and S2 should be to S6 and S7.

Line 330-332: It is mentioned that species richness was better explained than endemism, probably due to not having information about island age and intra-archipelago dynamics. A quick phrase or sentence to mention the difficulty of getting this information for each island would be helpful to justify why these were not included, for readers who may not be aware of the limited data availability for these variables.

Line 400: Small typo in "likelly"

Lines 23, 163, 190: The word "analyzes" should be spelled "analyses"

The scope of the dataset (number of islands and species) is mentioned in the methods, but could be highlighted even earlier if possible, such as the introduction or even the abstract. The dataset is quite a valuable contribution from this study, and would likely be of interest to many ecologists and conservation biologists who might skim the abstract or introduction.

Figure 4 is a great way of representing these complex results across realms, which can be difficult to communicate clearly. Well done!

Decision letter (RSPB-2021-1879.R0)

27-Sep-2021

Dear Dr Barreto:

Your manuscript has now been peer reviewed and the reviews have been assessed by an Associate Editor. The reviewers' comments (not including confidential comments to the Editor) and the comments from the Associate Editor are included at the end of this email for your reference. As you will see, the reviewers and the Editors have raised some concerns with your manuscript and we would like to invite you to revise your manuscript to address them.

Research ethics:

Use of animals and field studies:

It is a condition of publication that you make available the data and research materials supporting the results in the article. Please see our Data Sharing Policies (<https://royalsociety.org/journals/authors/author-guidelines/#data>). Datasets should be deposited in an appropriate publicly available repository and details of the associated accession number, link or DOI to the datasets must be included in the Data Accessibility section of the article (<https://royalsociety.org/journals/ethics-policies/data-sharing-mining/>). Reference(s) to datasets should also be included in the reference list of the article with DOIs (where available).

Please submit a copy of your revised paper within three weeks. If we do not hear from you within this time your manuscript will be rejected. If you are unable to meet this deadline please let us know as soon as possible, as we may be able to grant a short extension.

Best wishes,
Dr Locke Rowe
mailto: proceedingsb@royalsociety.org

Associate Editor
Board Member: 1
Comments to Author:

I congratulate the authors on a fascinating revisit of the insular biodiversity patterns. The role of environmental and climate data (e.g. climate change velocity) used to describe insular biodiversity is very interesting. I agree with the reviewers that much of the writing is clear, the sample sizes are adequate, and the analytical approaches are rigorous for the questions being asked.

I suggest that the authors consider the reviewers' suggestions to examine their explanations of pSIE, the role of biogeography in the analyses, and make the required changes. There are a few aspects that I will request the authors' comments on 1) the possible role of undescribed diversity: I can imagine that several tropical islands are not well explored or described for small mammal diversity
2) implications of the IUCN range maps for small islands - did these match island boundaries etc. I have experienced several cases where older EOO ranges included multiple islands, or the island size was incorrectly depicted by EOO. I note that you have excluded islands less than 1sqkm, but that was when comparing with environmental data.
3) the role of extinction in the outcomes of climate change (both area or velocity) from LGM. At present the discussion appears to be driven towards increased connectivity except for Line421.

I must also add that none of the above impact your analyses or conclusions. Still, even with data availability limitations, it would be good if the authors could comment on the potential impacts. I also request the authors to explain the derived climate variable (SLMP, GMMC, CCVT). The source of the data is cited, but an explanation of the variables and the expectation for using them in your analyses will help the reader. Lines 70-73 have some information on LGM impacts but an explanation of the main variables on similar lines will be useful.

I also agree with the suggestions of two reviewers that the dataset used here - species lists for each island, is a valuable contribution in itself and could be deposited in a repository for access to the community. At present, the Appendix 1 is island-level diversity metrics and the explanatory variables used in the models, but the species lists used to calculate these diversity metrics for each island are not included.

Reviewer(s)' Comments to Author:
Referee: 1
Comments to the Author(s)
General comments:

I have read the manuscript titled 'Area, isolation, and climate explain the diversity of mammals on island worldwide'. In this paper, the authors tested several key predictions of the island biogeography theory using mammal data worldwide. They found contrasting patterns between the insular diversity of bats and non-volant mammals and confirmed the leading role of island area and isolation in driving island biodiversity.

The paper was written in a succinct and logical manner. The results and the arguments are easy to follow. In general, I think the paper makes a nice contribution to the knowledge of island biodiversity worldwide. I have three general comments to further improve the paper:

1. I think the theoretical background of the study should be further developed in the introduction, especially regarding how pSIE is predicted by extinction and colonization rates. There is a nice theoretical paper on this subject, which I think the manuscript could benefit a great deal from:

Chen, X.-Y., & He, F. (2009). Speciation and endemism under the model of island biogeography. *Ecology*, 90(1), 39–45. doi:10.1890/08-1520.1

It might provide additional insights about why island area does not explain the pSIE (Fig. 3). This deviation from your prediction should also be given more attention in the discussion section.

2. I wonder if the current title is a little bit misleading and masks some important messages unique to the paper. For example, judging from table 2, biological realm explains much more variation of insular biodiversity than any other variables. This seems to suggest the dominant role of historical contingency/evolutionary history on island biodiversity. The current method treats the biological realm as random effects. However, I am curious to know if each biological realm is fitted by a different model with the same set of predictors, will the R² be different among biological realms? In other words, will climatic and insular variables have different explanatory powers in different regions?

3. More introductions should be provided addressing the prior expectations about the relative importance of climatic variables and island variables. In addition, the effects of climate change velocity and elevation heterogeneity should also be properly introduced before the method section.

Minor comments:

1. L61, lower extinction rate on larger island could also contribute to higher pSIE (Equation 5; Chen and He, 2009). But if the island species richness is fixed, lower extinction rate (larger islands) can actually lead to lower pSIE (Fig. 3; Chen and He, 2009)

Chen, X.-Y., & He, F. (2009). Speciation and endemism under the model of island biogeography. *Ecology*, 90(1), 39–45. doi:10.1890/08-1520.1

2. L82, 'and species numbers' is redundant.

3. L103, is there a reason that climatic effect is expected to be weaker than island area and isolation?

4. L186, not enough variation in what variable? What would be the random effect if a glmm is used?

5. L186-188, since pSIE is a proportion data. I think beta regression might be more suitable for this. The authors might consider doing a beta regression and put it in the supplementary (but this is totally up to the authors).

6. L191-193, I am not familiar with this approach. Could you give more explanations about why subtracting 1 from species richness improves the fitting of a negative binomial model? Also please add citations if this has been implemented in other studies.

7. L196-200, could you be more specific about which predictors have biological realm as a random effect for the slope (all the predictors or just some of them?).

8. L214, how many islands were excluded from the analysis? It should be mentioned in the method section.

9. Figure 1, 'ate least' should be 'at least'.

10. Figure 2, all species include bats and non-volant species. I suggest removing the 'all species' group as it is not independent of the other groups, a pairwise comparison in this case is therefore not appropriate.

11. L263, the tables start with table 2. There is no table 1 in the paper.

12. L263, table 2. Consider adding the results of the partitioning of variance of climatic variables and island variables.

13. L341, it is worthwhile to explain why area does not influence pSIE, which is contradictory to the prediction of Chen and He 2009.

14. Consider removing some tables in S2-S9. It seems unnecessary to include all the spatial-autocorrelation analysis.

Referee: 2

Comments to the Author(s)

In this manuscript, the authors examine the drivers of native mammal richness and endemism in islands globally. They perform analyses for all mammals, for bats only and for non-volant mammals. Geographical features (area and isolation) explain most of richness and endemism, whereas environmental drivers appear to be more idiosyncratic and their effects vary across biogeographic realms.

GENERAL COMMENTS

This is a nice, pretty straightforward paper that present convincing results. The manuscript covers all major pitfalls I could think of, in terms of data compilation and analyses. I only have some minor comments described below to clarify a few points.

The data available in Supplementary Information only contains the summary data per island (richness, SIE and pSIE), rather than the full list of species. It would be extremely useful to the scientific community to make the whole data available in a public repository, with a DOI so that it is citable and credit can be given to the authors of course.

SPECIFIC COMMENTS

l.31: Why is this result unexpected? Actually, important changes in climatic conditions might lead to speciation through adaptation, so it does not seem counterintuitive.

l.34: evidence on → evidence of

l.47: the full stop should be a comma, or the new sentence should be rephrased to be grammatically correct.

l.61: the relationship between area and endemism should be unpacked a bit. If I get the reasoning correctly, if a population gets isolated from the continent, it can evolve in a new species due to allopatric speciation, regardless of area, right? But the larger the island, the more opportunity for additional speciation events to occur, leading to more than potentially one new species?

l.149-151: which time period was used to compute the temperature and precipitation means and sd?

l.162: I do not get the meaning of the sentence "Due to the lower number of sample units per realm on the analyzes of pSIE". By "sample units", do you mean islands? Why are there less sample units for pSIE than for richness and SIE?

l.171: would it be possible to get the commented R script used for the analyses as supplementary information?

l.238: according to figure 2e, temperature variability was only significant when comparing all species and bats, not all species and non-volant species.

l.234-244: this section only refers to significance, but some differences are larger than others, and it would be good to mention effect size. For example, distributions were significantly different for area and temperature sd, but the differences were quite small compared to current isolation, for example.

Table 2 should be Table 1

l.251: According to Table 2, the range of conditional R2 is between 27% and 94%, not 27% and 90%.

l.259: Figs. 2a → Fig. 3a

l.291-294: I do not think you can directly compare absolute values of coefficients between models, since the response variables differ.

l.296: why was the interaction between area and isolation only used for richness, and not the other 2 models?

Figure 4: It would be useful to distinguish between significant and non-significant coefficients. Significant coefficients could be represented by filled disks, and non-significant coefficients by hollow rings, using the same colours for positive and negative values.

l.306: the colours in Figure 4 are not blue and red.

l.309-321: It could be useful to show the range of values for environmental variables in the different realms, and to discuss if similar trends in coefficient values can be associated with similar range of values (e.g. possibly positive and negative relationships for realms with lower and higher temperatures, respectively). Also, there could be interactions between environmental variables. For example, the relationship between richness and temperature may differ in dry and wet environment?

l.330: then → than

l.330: where do these values come from? There do not appear in Table 2.

l.340: The effects of area and isolation → Effects of area and isolation

l.358: “elements of the landscape” is unclear. Please be more specific, and give a couple of examples.

l.366: The flight capacity → Flight capacity

l.369: Name these different processes. I guess dispersal, environmental and speciation processes?

l.395: The effects of the environment → Effects of the environment

l.396 onwards: This section lacks some discussion on why the effects of climate vary so much with the realm.

l.410: is mostly from → mostly results from

l.411: And that → In addition,

l.412: it is needed elevation measures at finer resolutions → elevation measures at finer resolutions are needed.

l.414-422: I am not sure if the last explanation really makes sense here, as according to it, to be endemic in an island with high VCC, this one island must still have a lower VCC than multiple surrounding ones in which the same species would go locally extinct. It does not seem that this could generate a positive relationship. This explanation could also be tested by using the

difference of VCC in an island with the VCC in surrounding islands. Rather, my guess is that in the mainland, vagile mammal species can move to follow climate change, whereas they cannot on islands. To survive, they would need to adapt, which can lead to speciation.

l.429: associated to species attributes (i.e. flight capacity) → associated with species vagility/dispersal ability/dispersal mode. “Attributes” is too general, whereas is specifically refers to a specific attribute.

l.434: “For instance” does not really work here. It suggests an example of how using “more in-depth regional or clade-specific focus” would enable us to better understand the relationship between endemism and environmental drivers will follow, but instead moves on to generalize the insights from this study to other systems.

l.436: The link between this study and deforestation, which would focus on habitat loss and exclude speciation, is not clear and seems a bit far-fetched.

Referee: 3

Comments to the Author(s)

This is an ambitious attempt to uncover generalities in the drivers of mammal diversity on islands at the global scale. The authors built an extensive global dataset of mammal diversity on islands, and compiled an accompanying dataset of physical and climatic characteristics of these islands to test hypotheses grounded in island biogeography theory. They succeed in uncovering generalities in the drivers of diversity that have not yet been identified for mammals on islands at the global scale. Overall, the study presents these contributions in a way to interest a broad readership as well as readers with a specific interest in island biogeography and mammals.

This is an important contribution to the body of evidence supporting the importance of island area and isolation in shaping diversity. The dataset which was compiled for this study is also a valuable contribution to macro-ecological research, due to its scale, the inclusion of bats and endemic species, and the rigour with which it was built. The methodology for data collection, validation, and analysis is justified both ecologically and statistically throughout the study. The manuscript was well-written and had a good logical flow throughout the text, and was overall an interesting read. The introduction sets up the study on a solid foundation of theoretical concepts that are grounded in examples and knowledge of how mammals live on and disperse between islands. I have just a few questions and suggestions about points that deserve more clarification or elaboration to drive the study’s message home.

My main comment is about endemism, which is measured both as the number of single island endemics (SIE) and as the proportion of single island endemics (pSIE). I found myself wondering why both of these measures were investigated rather than just pSIE or vice-versa. What was the reasoning for including both? It could be helpful to explain this decision early in the text, especially if it is because they give different insights. The two metrics were also related quite differently to isolation in the results, but the reason for this should be discussed somewhere to guide the interpretation of this result. If there is not a clear justification for keeping both, I would consider focusing on one of the two (maybe pSIE) in the main text to have a single metric for richness and a single metric for endemism, which would simplify the manuscript.

The realm labels in Figure 1b) could be larger. They are difficult to read, especially on a smaller screen.

Line 181: I believe this reference to Figs S1 and S2 should be to S6 and S7.

Line 330-332: It is mentioned that species richness was better explained than endemism, probably due to not having information about island age and intra-archipelago dynamics. A quick phrase or sentence to mention the difficulty of getting this information for each island would be helpful

to justify why these were not included, for readers who may not be aware of the limited data availability for these variables.

Line 400: Small typo in "likelly"

Lines 23, 163, 190: The word "analyzes" should be spelled "analyses"

The scope of the dataset (number of islands and species) is mentioned in the methods, but could be highlighted even earlier if possible, such as the introduction or even the abstract. The dataset is quite a valuable contribution from this study, and would likely be of interest to many ecologists and conservation biologists who might skim the abstract or introduction.

Figure 4 is a great way of representing these complex results across realms, which can be difficult to communicate clearly. Well done!

Author's Response to Decision Letter for (RSPB-2021-1879.R0)

See Appendix A.

Decision letter (RSPB-2021-1879.R1)

15-Nov-2021

Dear Dr Barreto

I am pleased to inform you that your manuscript RSPB-2021-1879.R1 entitled "Area, isolation, and climate explain the diversity of mammals on island worldwide" has been accepted for publication in Proceedings B.

The referees and AE raised the issue of access to the data underlying that given in the supplementary materials. In response you suggested that broad access to these data would put at risk ongoing analyses associated with a PhD thesis. I am sympathetic to this concern. At the same time, you suggested the possibility of depositing the data with a 2yr embargo on access. I am strongly encouraging you to do so, but I am not making acceptance of the manuscript contingent on deposition of these data. It is for this reason that the decision is "accept with minor revision" rather than "accept as is". Please come to a decision on the deposition of these data, and if necessary revise your section on data accessibility.

Because the schedule for publication is very tight, it is a condition of publication that you submit the revised version of your manuscript within 7 days. If you do not think you will be able to meet this date please let us know.

When submitting your revised manuscript, you will be able to respond to the comments made by the referee(s) and upload a file "Response to Referees". You can use this to document any changes

you make to the original manuscript. We require a copy of the manuscript with revisions made since the previous version marked as 'tracked changes' to be included in the 'response to referees' document.

Sincerely,
Dr Locke Rowe
Editor, Proceedings B
<mailto:proceedingsb@royalsociety.org>

Associate Editor:

Board Member

Comments to Author:

Thank you, authors, for comprehensively addressing all the concerns of the referees.

Congratulations again for a fascinating study,

I appreciate the PhD goals of an author and accept the suggestion to deposit the data under a 2-year embargo.

Reviewer(s)' Comments to Author:

Author's Response to Decision Letter for (RSPB-2021-1879.R1)

See Appendix B.

Decision letter (RSPB-2021-1879.R2)

19-Nov-2021

Dear Dr Barreto

I am pleased to inform you that your manuscript entitled "Area, isolation, and climate explain the diversity of mammals on island worldwide" has been accepted for publication in Proceedings B.

Data Accessibility section

Open Access

Paper charges

Sincerely,

Appendix A

Editor's comments to author

I congratulate the authors on a fascinating revisit of the insular biodiversity patterns. The role of environmental and climate data (e.g. climate change velocity) used to describe insular biodiversity is very interesting. I agree with the reviewers that much of the writing is clear, the sample sizes are adequate, and the analytical approaches are rigorous for the questions being asked. I suggest that the authors consider the reviewers' suggestions to examine their explanations of pSIE, the role of biogeography in the analyses, and make the required changes.

Response:

Dear editor,

We appreciate the very constructive comments by you and the three reviewers and thank you for the opportunity to present this revised version of our manuscript. As stated in the cover letter, we agreed with most of the points raised and changed the manuscript accordingly. We removed pSIE from the main text (we moved all its associated results to the supplementary material) and we clarified the role of biogeography. In the following pages you will find our point-by-point responses to each of the comments.

There are a few aspects that I will request the authors' comments on:

1) the possible role of undescribed diversity: I can imagine that several tropical islands are not well explored or described for small mammal diversity.

2) implications of the IUCN range maps for small islands - did these match island boundaries etc. I have experienced several cases where older EOO ranges included multiple islands, or the island size was incorrectly depicted by EOO. I note that you have excluded islands less than 1sqkm, but that was when comparing with environmental data.

3) the role of extinction in the outcomes of climate change (both area or velocity) from LGM. At present the discussion appears to be driven towards increased connectivity except for Line 421.

I must also add that none of the above impact your analyses or conclusions. Still, even with data availability limitations, it would be good if the authors could comment on the potential impacts.

Response:

1: Many species are still unknown and even for those already described, we still likely do not know their complete geographical distribution (Linnean and Wallacean shortfalls, respectively). These limitations are common to most (if not all) macroecological studies and have the potential to bias the patterns and the inferences made from it. We expect that this limitation is further exacerbated on islands because they are often remote and studies in these regions tend to be expensive. However, we trust that these shortfalls did not drastically impact the conclusions of our study because of the great sample size (over 5,500 islands) and geographic extent (global), which are likely to dilute the effect of these limitations and allow the identification of a strong biological signal.

2: We did not find small islands to be the problem, at least not for mammals. In fact, we found a strong agreement between the species range maps and the island spatial polygons, even for small islands. As stated in the manuscript, inconsistencies in spatial overlap between both datasets were more frequent in regions with clusters of nearby islands and on islands near the continent shore (L. 128 - 132). Thus, we chose the conservative approach of exclude any of the islands for which we could not assign species with confidence.

We agree with the importance to comment on these two aspects and thus added the following statements to the discussion section (L. 363-371): *“We acknowledge that the study of current insular biodiversity patterns could potentially be impacted by the recent extinctions and range shifts due to human activities (e.g. [38,58]) and to the Linnean and Wallacean shortfalls [59]. For instance, a recent study shows that half of the mammal species on the well-known island of Luzon were unknown only a few years ago [60]. Despite these limitations, our findings are in line with the various expectations from island biogeography theory. That suggests that IUCN-derived diversity measures are robust to explore patterns of insular biodiversity [8], especially because of the large spatial scales and great number of sampling unities that can be derived from such dataset”*.

3: Extinction is likely to play a strong role in the outcomes of climate change since the LGM. We have made that clearer in the introduction by stating how increased connectivity would decrease extinction risk (L. 72) and how faster climatic changes would increase extinction risk (L. 77-79). We have also

expanded the discussion about climate change velocity (L. 451-464). We chose not to mention how changes in area with sea level fluctuation would impact mammal insular diversity because it is beyond the scope of this paper.

I also request the authors to explain the derived climate variable (SLMP, GMMC, CCVT). The source of the data is cited, but an explanation of the variables and the expectation for using them in your analyses will help the reader. Lines 70-73 have some information on LGM impacts but an explanation of the main variables on similar lines will be useful.

Response: We have added a full description of SLMP, GMMC and CCVT in the methods section. In it, we mentioned how SLMP and GMMC are proxies of current and historical isolation, respectively, which links back to the theory and to the expectations we detailed in the introduction. As for the CCVT, we explained in the methods section how it could be interpreted as a measure of the pace a species have to move to track climate and in the introduction, we set a parallel with findings from continental areas.

Introduction (L.66 - 79): *“Islands are not static over time and past geological and climatic conditions left strong imprints on current patterns of biodiversity, especially endemism [20,21]. (...) Abiotic variability associated with climate change since the LGM is also expected to have influenced current patterns of biodiversity [22]. If insular dynamics are like continental ones, we could expect faster changes in climate over space and time leading to lower endemism because species get extinct or shift its range by tracking climate [22,23]”.*

Methods section (L.170 - 184): *“SLMP is a proxy of island isolation with great predictive power and was calculated as the log10-transformed sum of the proportion of surrounding landmass within buffer distances of 100, 1,000 and 10,000 km around each island perimeter [44]. GMMC is a binary descriptor of historical isolation that uses past and present global bathymetry data to infer if islands were connected to the continent during the last glacial maximum by assuming the estimated sea level decrease of 122m at 18,000 years ago (more details in [1]). We multiplied SLMP by -1 and coded GMMC as 0 being connected and 1 being disconnected to the mainland during the LGM, so both metrics represent isolation (i.e., higher SLMP and GMMC represent greater isolation). Hereafter we will refer to those variables as “current isolation” and*

“past isolation”. CCVT over the past 21,000 years was calculated by dividing the difference in mean annual temperature between past and present by the spatial change in present mean temperature [1,45]. CCVT is interpreted as the speed at which the organism would have to move to keep pace with historical temperature change, assuming no change in topography [1,45]”.

I also agree with the suggestions of two reviewers that the dataset used here - species lists for each island, is a valuable contribution in itself and could be deposited in a repository for access to the community. At present, the Appendix 1 is island-level diversity metrics and the explanatory variables used in the models, but the species lists used to calculate these diversity metrics for each island are not included.

Response: We appreciate the suggestion, and we understand how important this data can be for the scientific community and even for us in terms of citation. However, we are conducting further studies using the presence and absence dataset and, thus, feel it is prudent not to make it available just yet. Some of these studies are part of PhD thesis that will still take some time to get published. Given that for the present paper we only used diversity measures derived from the species list (richness and endemism), they would be sufficient to guarantee the reproducibility. We hope this is not a problem, but if it is, we can deposit the data under a 2-year embargo.

Reviewer comments to author

Referee: 1

I have read the manuscript titled ‘Area, isolation, and climate explain the diversity of mammals on island worldwide’. In this paper, the authors tested several key predictions of the island biogeography theory using mammal data worldwide. They found contrasting patterns between the insular diversity of bats and non-volant mammals and confirmed the leading role of island area and isolation in driving island biodiversity.

The paper was written in a succinct and logical manner. The results and the arguments are easy to follow. In general, I think the paper makes a nice contribution to the knowledge of island biodiversity worldwide.

Response: We thank the reviewer for such nice comments about our paper and for all these valuable suggestions that substantially improved the manuscript.

Major comments:

I have three general comments to further improve the paper:

1. I think the theoretical background of the study should be further developed in the introduction, especially regarding how pSIE is predicted by extinction and colonization rates. There is a nice theoretical paper on this subject, which I think the manuscript could benefit a great deal from:

Chen, X.-Y., & He, F. (2009). Speciation and endemism under the model of island biogeography. Ecology, 90(1), 39–45. doi:10.1890/08-1520.1

It might provide additional insights about why island area does not explain the pSIE (Fig. 3). This deviation from your prediction should also be given more attention in the discussion section.

Response: Thank you for pointing out the paper from Chen & He (2009). We decided to remove pSIE from the main text, as all its results could be explained by the findings from richness and endemism, which are the variables used to derive pSIE (SIE/richness). For example, the lack of an area-pSIE association could result from area increasing both, richness and SIE at similar rates. In any case, the paper by Chen & He helped us to better interpret and discuss our results for richness and endemism: “*The contrasting relationships of island isolation with richness (negative) and endemism (positive) are consistent with the expected reduction in immigration and gene flow in more isolated islands [3,16,19,32] so that increases in diversity are mostly due to endemic species [17,32]*” (L. 387-390).

2. I wonder if the current title is a little bit misleading and masks some important messages unique to the paper. For example, judging from table 2, biological realm explains much more variation of insular biodiversity than any other variables. This seems to suggest the dominant role of historical contingency/evolutionary history on island biodiversity. The current method treats the biological realm as random effects. However, I am curious to know if each

biological realm is fitted by a different model with the same set of predictors, will the R² be different among biological realms? In other words, will climatic and insular variables have different explanatory powers in different regions?

Response: We appreciate the suggestion. We chose to include realm as a random effect because we know that historical contingency and evolutionary history are likely play a strong role in structuring global patterns (which we found to be the case here) and ignoring it would be misleading. However, our main goal is to accommodate this source of variance to look past it and be able to identify generalities on how island characteristics are associated with biodiversity. It is not in the scope of the present paper but is in our interest to dissect the particularities of each realm in a follow up study. We have edited a sentence in the introduction to make it clearer why we included realm as a random effect (L. 109-112): “*We seek to establish the relative importance of island characteristics as predictors of richness and endemism of bats and non-volant mammals, while accommodating the variation among biogeographical realms due to the deep historical factors [33].*”. And we added to the discussion a sentence highlighting the effects of historical contingency (L. 351-354): “*Overall, species richness was better explained than endemism (mean R² of 89% vs 55%, respectively; Table 1) because we were able to statistically accommodate biogeographic realms as a random effect. This result suggests a strong role of historical contingency as a source of variation in insular biodiversity*”.

3. More introductions should be provided addressing the prior expectations about the relative importance of climatic variables and island variables. In addition, the effects of climate change velocity and elevation heterogeneity should also be properly introduced before the method section.

Response: We agree.

We split the introduction paragraph that refers to climate so that the first part is about direction of the effect and the relative importance compared to the physical characteristics of the islands (L. 80 - 87) and the second part is about heterogeneity (L. 88 - 96). At the end of the introduction, where we listed our expectations, we now explain why we expect physical conditions to

have stronger effects than the climate (L. 118-121): “*the direction of the effect of climate on insular biodiversity is similar to that found on continents, but effect will be weaker than those of island’s physical conditions because area and isolation simultaneously affect the three ultimate causes of biodiversity patterns: speciation, extinction and dispersal*”.

We have included in the introduction the expectations for the effects of climate change velocity (L. 66 – 79): “*Islands are not static over time and past geological and climatic conditions left strong imprints on current patterns of biodiversity, especially endemism [20,21]. (...) Abiotic variability associated with climate change since the LGM is also expected to have influenced current patterns of biodiversity [22]. If insular dynamics are like continental ones, we could expect faster changes in climate over space and time leading to lower endemism because species get extinct or shift its range by tracking climate [22,23]*”.

Our expectation for how elevation heterogeneity influence mammalian biodiversity is lumped with the expectation from other intra-island heterogeneities and detailed on lines 88 to 96. We have now added a sentence about our expectations for these relationships: “*Variation in species richness and endemism also arise from the effect of intra-island heterogeneity, either in habitat, **topography** and/or climate, which tends to be greater on larger islands [26,27]. More heterogeneous environments facilitate the co-existence of a wide range of species with different environmental requirements through niche partitioning [28,29]. Additionally, over evolutionary time, greater heterogeneity may increase speciation rates as a result of niche shifts, ecological specialization and increased reproductive isolation [29,30]. Thus, a positive association is expected between **topographic** and climate heterogeneity and both, species richness and endemism*”.

Minor comments:

L61, lower extinction rate on larger island could also contribute to higher pSIE (Equation 5; Chen and He, 2009). But if the island species richness is fixed, lower extinction rate (larger islands) can actually lead to lower pSIE (Fig. 3; Chen and He, 2009).

Chen, X.-Y., & He, F. (2009). Speciation and endemism under the model of island biogeography. *Ecology*, 90(1), 39–45. doi:10.1890/08-1520.1

Response: Our sentence was unclear and led to confusion. In the new version of the paper we chose to remove pSIE from the main text, so the sentence is no longer included.

L103, is there a reason that climatic effect is expected to be weaker than island area and isolation?

Response: We have now made it clear why we expect stronger effects of the physical characteristics of the islands. The text now reads: “*the direction of the effect of climate on insular biodiversity is similar to that found on continents, but effect will be weaker than those of island’s physical conditions because area and isolation simultaneously affect the three ultimate causes of biodiversity patterns: speciation, extinction and dispersal*” (L. 118-121).

L186, not enough variation in what variable? What would be the random effect if a glmm is used?

Response: There was not enough variation in SIE. We have now made it clearer by stating that “*SIE was modeled using GLM because most islands contain only one or two endemics, and therefore, there was not enough variation in this variable to fit a GLMM*” (L. 209 – 211). If it was statistically possible to use a GLMM we would follow the same settings as for the other response variables and use realm as random effect.

L186-188, since pSIE is a proportion data. I think beta regression might be more suitable for this. The authors might consider doing a beta regression and put it in the supplementary (but this is totally up to the authors).

Response: We appreciate the suggestion. However, modeling pSIE using a binomial error distribution with species richness as prior weights – as we did – is as good an assessment as beta regression would be. In addition, it has the advantage of including realm as random effect. In any case, we decided to remove pSIE from the main text, as it offered no additional insight into the possible processes driving insular biodiversity.

L191-193, I am not familiar with this approach. Could you give more explanations about why subtracting 1 from species richness improves the fitting of a negative binomial model? Also please add citations if this has been implemented in other studies.

Response: Our dataset contains no island with zero species richness because we could not confidently score true absences, meaning that we have a zero-truncated distribution. Negative binomial models would not be able to properly fit to the data because it always predicts zeros (see figure 1 above). Other distribution models such as the Poisson were unfeasible because there was a clear overdispersion problem. A simple trick to circumvent this situation was to subtract one from all richness values and then proceed using zero-inflated models. This solution was proposed by an expert on the subject, Dr. Florian Artig, here: <https://github.com/florianhartig/DHARMA/issues/131>

We edited a sentence in the manuscript to improve clarity (L. 217 – 220): “*As our dataset contains no island with zero species, we subtracted one (1) from the species richness and from the number of endemics when modeling the biodiversity of all mammal species to improve the fit of negative binomial models because these models are designed to predict zeros*”.

Figure 1 – Residual plots and zero-inflation test from DHARMA R package showing a poor fit of the GLMM for species richness when using the negative binomial distribution for a dataset with no zeros.

L196-200, could you be more specific about which predictors have biological realm as a random effect for the slope (all the predictors or just some of them?)..

Response: We included random effects for the slopes all predictors. We have now made it clearer in the text (L. 223 – 225): “*We fitted GLMMs for species richness using realm as random effect to enable the estimation of different intercept and slopes for each realm across all the predictors, as this reduces type I error when compared to models with only random intercept*”.

L214, how many islands were excluded from the analysis? It should be mentioned in the method section.

Response: We have mentioned on line 189 to 191 that our final dataset comprised 5,592 islands and now added that information that there is an estimated total of about 17,000 islands worldwide: “*The final dataset comprised 5,592 islands (out of ~17,000 islands larger than 1km² worldwide [1]) of which 123 contained SIE (Fig. 1, Appendix 1)*”.

L82, ‘and species numbers’ is redundant.

Figure 1, ‘ate least’ should be ‘at least’.

L263, the tables start with table 2. There is no table 1 in the paper

Response: Thank you for the careful read of the manuscript. We corrected all these small mistakes.

Figure 2, all species include bats and non-volant species. I suggest removing the ‘all species’ group as it is not independent of the other groups, a pairwise comparison in this case is therefore not appropriate.

Response: The three groups compared in our ANOVA tests and illustrated in Figure 2 are independent. They are non-overlapping subsets of the complete dataset. The “all species” group includes islands where at least one bat and one non-volant mammal occur, whereas the “only bats” group and the “only non-volants” group include islands where the only species to occur there are bats and non-volants, respectively. We edited the methods section and the

figure legend to make it clearer by mentioning how many islands are in each group:

Methods (L. 200 – 203): “*To explore broad biodiversity patterns, we tested if islands that harbor only bats ($n = 1,831$), only non-volant species ($n = 2,094$), or representants from both groups ($n = 1,667$), tend to differ in latitude, physical and environmental characteristics using ANOVA tests (Figs. S5-6)*”.

Figure legend (L. 271-272): “*Comparison among islands that harbor only bats ($n = 1,831$; green), only non-volant mammals ($n = 2,094$; yellow), or both ($n = 1,667$; purple) (...)*”

L263, table 2. Consider adding the results of the partitioning of variance of climatic variables and island variables.

Response: We have not statistically partitioned the variance of climatic vs physical variables. Instead, we compared their relative importance by looking at their effect sizes (i.e., standardized slope coefficients). This is reported in Figure 3.

L341, it is worthwhile to explain why area does not influence pSIE, which is contradictory to the prediction of Chen and He 2009.

Response: We have chosen to remove pSIE from the main text, based on comments from other reviewers. The lack of an area-pSIE association could result from area increasing both, richness and SIE at similar rates.

Consider removing some tables in S2-S9. It seems unnecessary to include all the spatial-autocorrelation analysis.

Response: We appreciate the suggestion, but we think it is important to report in the supplementary material the spatial-autocorrelation analysis for all the nine regression models of our study, as they are not the same.

Reviewer comments to author

Referee: 2

General comments:

In this manuscript, the authors examine the drivers of native mammal richness and endemism in islands globally. They perform analyses for all mammals, for bats only and for non-volant mammals. Geographical features (area and isolation) explain most of richness and endemism, whereas environmental drivers appear to be more idiosyncratic and their effects vary across biogeographic realms.

This is a nice, pretty straightforward paper that present convincing results. The manuscript covers all major pitfalls I could think of, in terms of data compilation and analyses. I only have some minor comments described below to clarify a few points.

Response: We appreciate the thoughtful review and thank you for such positive feedback.

The data available in Supplementary Information only contains the summary data per island (richness, SIE and pSIE), rather than the full list of species. It would be extremely useful to the scientific community to make the whole data available in a public repository, with a DOI so that it is citable and credit can be given to the authors of course.

Response: We acknowledge the relevance of making the presence and absence data available and appreciate the suggestion for it to be citable. However, we are working on other publications using this data, and therefore, are not ready to make this data available just yet. For now, we prefer to make available the data used in the present study (i.e., species richness and endemism per island).

Specific comments:

l.31: Why is this result unexpected? Actually, important changes in climatic conditions might lead to speciation through adaptation, so it does not seem counterintuitive.

l.414-422: I am not sure if the last explanation really makes sense here, as according to it, to be endemic in an island with high VCC, this one island must still have a lower VCC than multiple surrounding ones in which the same species would go locally extinct. It does not seem that this could generate a positive relationship. This explanation could also be tested by using the difference of VCC in an island with the VCC in surrounding islands. Rather, my

guess is that in the mainland, vagile mammal species can move to follow climate change, whereas they cannot on islands. To survive, they would need to adapt, which can lead to speciation.

Response: Agreed. The positive association between climate change velocity and endemism is unexpected given what is known for continental areas, where this association is consistently negative. But you are correct to point out that species may also adapt to the new climatic conditions, which could spur speciation, especially when considering that we measure endemism as the number of single island endemics. It may be the case that local populations adapt and diverge from populations that occur on other islands, resulting in an increase in the number of single island endemics. We removed the “unexpectedly” from the abstract and expanded the discussion about the possible underlying causes of the positive association between climate change velocity and endemism (L. 450-463):

“The positive association between endemism and climate velocity since the LGM contrasts with findings for endemism on the mainland [23] and challenges the expectation that climate instability would favor the occurrence of generalist broad-ranged species [73]. On the continent, changes in temperature since the LGM influenced mammalian biodiversity through range shifts and increased probability of extinction [73]. On islands, where boundaries are hard, species may be less likely to shift their ranges to track new climates and may be under strong pressure to adapt to novel conditions. Adaptation of island organisms can quickly lead to divergence from populations of other islands and mainland [74] and to speciation events that culminate in an increase in single island endemics. Also, endemism might increase with climatic instability if environmental changes were comparatively milder on the focal island than on the mainland, which could be the case given that the ocean mass buffers climatic change on islands [75]. In this case, species that went extinct on the continent might have persisted on the island.”

l.34: evidence on → evidence of

l.47: the full stop should be a comma, or the new sentence should be rephrased to be grammatically correct.

Table 2 should be Table 1

l.259: Figs. 2a → Fig. 3a

l.306: the colours in Figure 4 are not blue and red.

l.330: then → than

l.340: The effects of area and isolation → Effects of area and isolation

l.366: The flight capacity → Flight capacity

l.395: The effects of the environment → Effects of the environment

l.410: is mostly from → mostly results from

l.411: And that → In addition,

l.412: it is needed elevation measures at finer resolutions → elevation measures at finer resolutions are needed.

l.429: associated to species attributes (i.e. flight capacity) → associated with species vagility/dispersal ability/dispersal mode. “Attributes” is too general, whereas is specifically refers to a specific attribute.

l.251: According to Table 2, the range of conditional R2 is between 27% and 94%, not 27% and 90%.

Response: Thank you for the careful read of the manuscript. We corrected all these small mistakes.

l.61: the relationship between area and endemism should be unpacked a bit. If I get the reasoning correctly, if a population gets isolated from the continent, it can evolve in a new species due to allopatric speciation, regardless of area, right? But the larger the island, the more opportunity for additional speciation events to occur, leading to more than potentially one new species?

Response: Correct. We have given more details about it in the introduction (L. 60-63): “*Thus, the rate at which new endemic species originate increases with isolation and area because of the greater chance for allopatric speciation in response to low migration and gene flow with the additional chance for intra-island speciation due to area-effects [3,17]*”.

l.149-151: which time period was used to compute the temperature and precipitation means and sd?

Response: We obtained temperature and precipitation annual mean and sd for each island by using the bioclimatic data from CHELSA climatologies, which is calculated through the monthly estimates across the years 1979 to 2013. This information was now added to the manuscript (L. 163-165): “*We derived temperature and precipitation data from CHELSA using monthly estimates across the years 1979 to 2013 [42]*”.

l.162: I do not get the meaning of the sentence “Due to the lower number of sample units per realm on the analyzes of pSIE”. By “sample units”, do you mean islands? Why are there less sample units for pSIE than for richness and SIE?

Response: Yes, we mean that there are few islands per realm with pSIE information. That happens because we only included islands with at least one endemic species, totaling 123 islands, whereas for we had 5,592 islands with at least one species for the richness analysis. The number of islands with SIE and pSIE is the same, but for SIE we didn’t use a mixed model with realm as random effect because of the lack of variability. This sentence was now removed from the main text and moved to the supplementary material along with all results from pSIE.

l.171: would it be possible to get the commented R script used for the analyses as supplementary information?

Response: Sure. The commented R script is now provided as supplementary information.

l.238: according to figure 2e, temperature variability was only significant when comparing all species and bats, not all species and non-volant species.

Response: Thank you for pointing that out. We have simplified the sentence, so it now refers more broadly to environmental conditions instead of highlighting each individual variable, for which there were some exceptions to the rule (L. 261-263): “*Compared to islands where either bats or non-volant*

*mammals occur, islands where both groups co-occur were larger, less isolated and had greater spatial variation in **environmental conditions***".

l.234-244: this section only refers to significance, but some differences are larger than others, and it would be good to mention effect size. For example, distributions were significantly different for area and temperature sd, but the differences were quite small compared to current isolation, for example.

Response: Agreed. However, there are too many comparisons for us to report effect sizes in a clear manner and that is why we chose to illustrate with the box plots in figure 2. We have added adverbs to highlight when differences are large (L. 263-265): "*Islands occupied only by bats had the opposite characteristics, **were considerably more isolated** and were also warmer and wetter (Figures 2 and S6)*" and "*Islands where only non-volant mammals occur (...) were **colder** and located at higher latitudes (...)*" (L. 265 – 268).

l.291-294: I do not think you can directly compare absolute values of coefficients between models, since the response variables differ.

Response: We appreciate the concern, but we are not aware of a statistical impediment to the comparison of standardized beta coefficients (i.e., standardized slope) between models with different response variables. The only objection we can think of is that there is a possibility that the coefficients are greater for species richness than for SIE, because richness is always greater than or equal to the number of endemic species, so changes in one unit of any predictor could be associated with greater changes in richness than in endemism. However, this is not what we observed in any comparison, which suggests that there is indeed a biological phenomenon being captured in the standardized beta coefficients that allows us to compare them.

l.296: why was the interaction between area and isolation only used for richness, and not the other 2 models?

Response: Because only for species richness we have enough sampling units (more than 5,500 islands) to estimate so many parameters.

Figure 4: It would be useful to distinguish between significant and non-significant coefficients. Significant coefficients could be represented by filled disks, and non-significant coefficients by hollow rings, using the same colours for positive and negative values.

Response: As far as we know, there is currently no way to calculate the significance of the estimated coefficients for each level of the random slope effect.

l.309-321: It could be useful to show the range of values for environmental variables in the different realms, and to discuss if similar trends in coefficient values can be associated with similar range of values (e.g. possibly positive and negative relationships for realms with lower and higher temperatures, respectively). Also, there could be interactions between environmental variables. For example, the relationship between richness and temperature may differ in dry and wet environment?

Response: We added a table with the range of values for each predictor within each realm in the supplementary files (Table S1) and the histogram of all these variables in appendix 2. There is no clear pattern of similar trends in coefficients being associated with similar range values or interactions between environmental variables. That can be already anticipated by the observation that there is considerable variation in the direction of association (negative and positive) between non-volants and bats within a same realm (Figure 4).

l.330: where do these values come from? There do not appear in Table 2.

Response: We are referring to the mean R^2 across the three models fitted for each diversity measure (i.e., all mammals, bats and non-volants). We have now made that clear by adding the mean to Table 1 (L. 295) and by editing the text (L. 350-351): “Overall, species richness was better explained *than* endemism (*mean R^2 of 89% vs 55%, respectively, Table 1*)”.

l.358: “elements of the landscape” is unclear. Please be more specific, and give a couple of examples.

Response: We clarified what we mean by “elements of the landscape” and provided examples (L. 389-393): “*Measures of island isolation at global scale should not focus solely on the distance to the mainland, but ideally,*

incorporate elements of the landscape, such as stepping stone distances or information on the surrounding landmasses [59]. That is especially the case for mammals, whose neighboring islands are known to be a more important source of colonizers than the closest mainland [11]".

l.369: Name these different processes. I guess dispersal, environmental and speciation processes?

Response: We now specify the processes (L. 400-403): *"The strength of the relationship between area and isolation with biodiversity varied with the group's vagility and the type of diversity measured, and is likely to reflect the relative importance of different processes, namely dispersal, extinction, and speciation"*.

l.396 onwards: This section lacks some discussion on why the effects of climate vary so much with the realm.

Response: We expanded our discussion of the possible causes of variation in environmental-diversity relationships across realms (L. 435-441): *"Environmental factors are more strongly associated with endemism than with richness and the intensity and direction of effect varied considerably across realms suggesting that environment-diversity relationships on islands are context-dependent and contingent to regional/archipelagic historical effects [57]. Specifically, variation in environmental-diversity relationships could be the result of different mechanisms prevailing in each realm or of the specific adaptations that each species pool acquired by evolving under different environmental gradients"*.

l.434: "For instance" does not really work here. It suggests an example of how using "more in-depth regional or clade-specific focus" would enable us to better understand the relationship between endemism and environmental drivers will follow, but instead moves on to generalize the insights from this study to other systems.

l.436: The link between this study and deforestation, which would focus on habitat loss and exclude speciation, is not clear and seems a bit far-fetched.

Response: Agreed. We have removed the sentence about generalizing the insights to deforestation sites.

Reviewer comments to author

Referee: 3

This is an ambitious attempt to uncover generalities in the drivers of mammal diversity on islands at the global scale. The authors built an extensive global dataset of mammal diversity on islands, and compiled an accompanying dataset of physical and climatic characteristics of these islands to test hypotheses grounded in island biogeography theory. They succeed in uncovering generalities in the drivers of diversity that have not yet been identified for mammals on islands at the global scale. Overall, the study presents these contributions in a way to interest a broad readership as well as readers with a specific interest in island biogeography and mammals.

This is an important contribution to the body of evidence supporting the importance of island area and isolation in shaping diversity. The dataset which was compiled for this study is also a valuable contribution to macro-ecological research, due to its scale, the inclusion of bats and endemic species, and the rigour with which it was built. The methodology for data collection, validation, and analysis is justified both ecologically and statistically throughout the study. The manuscript was well-written and had a good logical flow throughout the text, and was overall an interesting read. The introduction sets up the study on a solid foundation of theoretical concepts that are grounded in examples and knowledge of how mammals live on and disperse between islands. I have just a few questions and suggestions about points that deserve more clarification or elaboration to drive the study's message home.

Response: We appreciate this very positive feedback and thank you for the constructive review.

My main comment is about endemism, which is measured both as the number of single island endemics (SIE) and as the proportion of single island endemics (pSIE). I found myself wondering why both of these measures were investigated rather than just pSIE or vice-versa. What was the reasoning for including both? It could be helpful to explain this decision early in the text, especially if it is because they give different insights. The two metrics were also related quite differently to isolation in the results, but the reason for this should be discussed

somewhere to guide the interpretation of this result. If there is not a clear justification for keeping both, I would consider focusing on one of the two (maybe pSIE) in the main text to have a single metric for richness and a single metric for endemism, which would simplify the manuscript.

Response: That's a good point. Our initial idea was to use only pSIE but we found it more intuitive and even more interesting to interpret the results for richness and endemism separately. We realize now that the paper is more straightforward without pSIE in the main text.

The realm labels in Figure 1b) could be larger. They are difficult to read, especially on a smaller screen.

Response: Done.

Line 181: I believe this reference to Figs S1 and S2 should be to S6 and S7.

Line 400: Small typo in "likelly"

Lines 23, 163, 190: The word "analyzes" should be spelled "analyses"

Response: We corrected this.

Line 330-332: It is mentioned that species richness was better explained than endemism, probably due to not having information about island age and intra-archipelago dynamics. A quick phrase or sentence to mention the difficulty of getting this information for each island would be helpful to justify why these were not included, for readers who may not be aware of the limited data availability for these variables.

Response: We edited the sentence, which now reads (L. 353-355): "*Also, endemism is likely to be strongly influenced by island age [31] and intra-archipelagic dynamics [57], information that is still unknown for most islands*".

The scope of the dataset (number of islands and species) is mentioned in the methods, but could be highlighted even earlier if possible, such as the introduction or even the abstract. The dataset is quite a valuable contribution from this study, and would likely be of interest to many ecologists and conservation biologists who might skim the abstract or introduction.

Response: We have included this information earlier on. In the abstract: “*We evaluated the importance of physical, environmental, and historical factors on mammal richness and endemism across 5,592 islands worldwide. (...). Richness on islands ranged from one to 234 species, with up to 177 single island endemics*”. And in the introduction (L. 106-108): “*Here we compiled a unique dataset of mammal composition on 5,592 islands worldwide to investigate how mammalian richness and endemism relate to island attributes*”.

Figure 4 is a great way of representing these complex results across realms, which can be difficult to communicate clearly. Well done!

Response: Thank you for the very positive comment. We are glad to know that we were successful in visually communicating such complex results.

Appendix B

Dear Editor,

We appreciate the acceptance of our paper entitled “*Area, isolation, and climate explain the diversity of mammals on island worldwide*” by Barreto, E.; Rangel, T.F.; Pellissier, L. & Graham, C.H. We have now included the presence and absence matrix in Dryad (doi:10.5061/dryad.hmgqnk9j2). Thank you for agreeing with the 2-year embargo. Following the suggestions of the Dryad data curator, we will keep the submission in “Private for Peer Review” during these 2 years (copied email below). All other supplementary material, including the full database of species richness, endemism and physical and environmental characteristics of the islands are uploaded in the paper’s supplementary material and will be made available with the publication.

Dear Elisa Barreto,

Thank you for your inquiry.

I suggest keeping your submission in "Private for Peer Review" status until you are prepared to release your data. While in this status, your submission will not enter our curation process or proceed to publication until you notify us that you'd like to proceed or deselect the 'Private for Peer Review' checkbox (on page 3 of the submission form). This feature will provide you with full control over when your dataset should be made public. In the meantime, you'll receive automated system reminders regarding the status of your submission which you can simply ignore.

Thanks in advance for your efforts. By offering a clear description as part of your data publication, you are promoting efficiency, transparency, and reproducibility in the research community.

If you have any additional questions or concerns, please let us know. And, thank you for choosing Dryad to host your data.

Kind regards,

Jess

*Jessica Herzog
Data Curator
helpdesk@datadryad.org*

Sincerely,

Elisa Barreto

On behalf of all the authors.